# Effect of Dipeptidyl Peptidase-4 Inhibitors on Bone Health in Patients with Type 2 Diabetes Mellitus

**DOI:** 10.3390/jcm10204775

**Published:** 2021-10-18

**Authors:** Dong-Hwa Lee, Kyong Young Kim, Min Young Yoo, Hansol Moon, Eu Jeong Ku, Tae Keun Oh, Hyun Jeong Jeon

**Affiliations:** 1Department of Internal Medicine, Chungbuk National University College of Medicine and Chungbuk National University Hospital, Cheongju 28644, Korea; roroko@hanmail.net (D.-H.L.); eujeong.ku@gmail.com (E.J.K.); tgohkjs@chungbuk.ac.kr (T.K.O.); 2Department of Internal Medicine, Gyeongsang National University Changwon Hospital, Changwon 51472, Korea; kkylucky@gmail.com; 3Department of Nuclear Medicine, Chungbuk National University Hospital, Cheongju 28644, Korea; ckitten@naver.com (M.Y.Y.); sprtual@cbnuh.or.kr (H.M.)

**Keywords:** dipeptidyl peptidease-4 inhibitor, bone mineral density, trabecular bone score, type 2 diabetes mellitus

## Abstract

Patients with type 2 diabetes (T2DM) have a higher risk of bone fracture even when bone mineral density (BMD) values are normal. The trabecular bone score (TBS) was recently developed and used for evaluating bone strength in various diseases. We investigated the effect of DPP-4 inhibitors on bone health using TBS in patients with T2DM. This was a single-center, retrospective case-control study of 200 patients with T2DM. Patients were divided into two groups according to whether they were administered a DPP-4 inhibitor (DPP-4 inhibitor group vs. control group). Parameters related to bone health, including BMD, TBS, and serum markers of calcium homeostasis, were assessed at baseline and after one year of treatment. We found TBS values increased in the DPP-4 group and decreased in the control, indicating a significant difference in delta change between them. The BMD increased in both groups, with no significant differences in delta change between the two groups observed. Serum calcium and 25-hydroxy vitamin D3 increased only in the DPP-4 inhibitor group, while other glycemic parameters did not show significant differences between the two groups. Treatment with DPP-4 inhibitors was associated with favorable effects on bone health evaluated by TBS in patients with T2DM.

## 1. Introduction

Patients with type 2 diabetes mellitus (T2DM) have increased risk of several diabetic complications. Common complications include vascular conditions such as coronary heart disease, cerebrovascular disease, nephropathy, retinopathy, and peripheral artery disease [1]. In addition, hyperglycemia can alter bone metabolism and lead to increased risk of osteoporosis and fracture in patients with T2DM [2]. Osteoporosis and osteoporotic fractures are strongly associated with morbidity and mortality; therefore, bony complications need to be considered during management of T2DM. The effects of antihyperglycemic agents on bone metabolism have been investigated in several previous studies. For example, thiazolidinedione showed a significant association with the reduction in bone mineral density (BMD) and increased the risk of fractures [3]. Other studies have reported that metformin, sulfonylurea, and insulin did not impair bone metabolism [4,5].

Dipeptidyl peptidase-4 (DPP-4) inhibitors are a class of antihyperglycemic drugs that are included in incretin-based therapy. In Korea, the use of DPP-4 inhibitor has dramatically increased after its introduction as a new class of antihyperglycemic agents, which now make up one-third of the market share [6]. Increatin-based therapies benefit bone metabolism by increasing the glucagon-like peptide-1 hormone [7]. However, studies of the association between DPP-4 inhibitors and fracture have shown inconsistent results [8,9,10].

BMD measurements by dual energy bone densitometry (DXA) is used as a standard method for assessing osteoporosis and risk of fracture. Although it is well-known that patients with diabetes have an increased risk of osteoporotic fracture, BMD is higher in patients with T2DM than in those without diabetes [11]. Therefore, BMD alone may be insufficient to predict the development of osteoporosis and fractures in patients with T2DM, and alternative methods are needed to evaluate these risks in these patients. The trabecular bone score (TBS) has been developed to estimate bone quality through the assessment of bone microarchitecture based on the lumbar spine DXA [12]. A previous study demonstrated that TBS was a good predictive value for fracture risk independent of BMD [13]. In another study, TBS was lower in patients with T2DM than normal controls, indicating that TBS can be a useful predictor for bone health in patients with T2DM [14]. Currently, the effect of DPP-4 inhibitors on bone health has not been fully elucidated. Moreover, studies evaluating the bone quality while using DPP-4 inhibitors are lacking. The aim of this study was to evaluate the potential benefits of DPP-4 inhibitors on bone health using TBS in patients with T2DM.

## 2. Materials and Methods

### 2.1. Study Population and Design

This was a single-center, case-control study retrospectively conducted at the Chungbuk National University Hospital (CBNUH). The data was collected through a review of medical records from January 2014 to December 2020. The eligible patients were included according to the following criteria: (1) patients were diagnosed with T2DM and treated at the CBNUH, and (2) patients who had DXA performed more than two times during the study period. Eligible patients were divided into two groups that were or were not administered a DPP-4 inhibitor. Patients who used a DPP-4 inhibitor for less than 6 months during the observation period were excluded from the DPP-4 inhibitor group. For the control group, patients who took a DPP-4 inhibitor at least once during the observation period were excluded. In both groups, we excluded patients if the date between their two DXA examinations were more than one year. Patients with insufficient medical records were also excluded from the analysis. Each subject underwent anthropometric assessment and laboratory tests according to a schedule routinely performed in patients with T2DM. This study was approved by the relevant ethics committees (The Institutional Review Board at Chungbuk National University Hospital, approval No. 2019-04-018-001) and conducted in accordance with the Declaration of Helsinki.

### 2.2. Measurement of Anthropometric and Biochemical Variables

Height (cm) and body weight (kg) were measured at the time of DXA by standard protocols to the nearest 0.1 cm and 0.1 kg, respectively. The body mass index (BMI) was calculated by dividing the body weight by the square of the height (kg/m^2^). Blood samples were performed in the morning after a 12-h fast that included any medication. The glycated hemoglobin (HbA1c) level was measured by high-performance liquid chromatography on an ADAMS™ A1c HA-8180T (ARKRAY, Inc., Kyoto, Japan). Fasting plasma concentrations of total cholesterol, triglycerides, high-density lipoprotein (HDL) cholesterol, low-density lipoprotein (LDL) cholesterol, serum creatinine, aspartate aminotransferase (AST), and alanine aminotransferase (ALT) were measured on a TBA-FX8 chemistry analyzer (Toshiba, Tokyo, Japan). 25-hydroxy vitamin D3 was measured using Alinity i (Abbott Diagnostics, Lake Forest, IL, USA).

### 2.3. Measurements of Bone Mineral Density and Trabecular Bone Score

Measurements of BMD were made by DXA (GE Lunar Prodigy Advance, Lunar Corporation, General Electric, Madison, WI, USA) at three skeletal sites (lumbar spine, femoral neck, and total hip) in all subjects at baseline and follow-up (one year). The same DXA instrument was used for both measurements. Analysis of the DXA date was performed with enCORE Software version 2005 9.30.044 (GE Healthcare, Madison, WI, USA) in accordance with the manufacturer’s recommendations. The L1-4 value was included in the analysis for BMD and TBS. The TBS was calculated using the TBS iNsight Software, version 3.02 (Med-Imaps, Pessac, France) from DXA images of the same vertebrae as the BMD measurements.

### 2.4. Statistical Analysis

Continuous data are expressed as the mean ± SD values. Categorical data are reported as percentages (%). The baseline characteristics were compared using the Student’s *t*-test for continuous variables or the chi-square test for categorical variables. A paired *t*-test was used to evaluate changes between the baseline and the follow-up. A *p*-value < 0.05 was considered statistically significant. All statistical analyses were performed using SPSS for Windows software 22.0 (IBM Corp., Armonk, NY, USA).

## 3. Results

### 3.1. Baseline Characteristics of the Study Subjects

A total of 200 patients with T2DM (*n* = 100 in each group) were included in the analysis. The demographic and clinical characteristics of these subjects are shown in Table 1. The mean age and BMI of the patients were 67.7 ± 10.5 years and 25.5 ± 3.8 kg/m^2^ in the DPP-4 inhibitor group and 68.1 ± 11.1 years and 24.4 ± 4.4 kg/m^2^ in the control group (*p* = 0.824 and 0.087), respectively. The number of women was similar in both groups (90 women in the DPP-4 inhibitor group vs. 88 women in the control group, *p* = 0.822). The mean duration of DM was 9.4 ± 9.1 years in the DPP-4 inhibitor group and 8.5 ± 9.1 years in the control group (*p* = 0.551). The mean HbA1c was significantly higher in the DPP-4 inhibitor group than in the control group (7.5 ± 1.4% vs. 6.7 ± 1.1%, *p* = 0.001). However, the fasting plasma glucose was not significantly different between the two groups (139.9 ± 39.8 mg/dL vs. 132.2 ± 36.8 mg/dL, *p* = 0.454). Other laboratory parameters, including lipid and bone metabolism, liver function, and kidney function, showed similar results in both groups.

There were some differences observed between the types of medications prescribed at baseline in the two groups. In the DPP-4 inhibitor group, the number of patients prescribed sulfonylurea and metformin was significantly larger than in the control group (31.0% vs. 10.0%, *p* < 0.001 and 81.0% vs. 62.0%, *p* = 0.004, respectively). In contrast, the proportion of insulin users was higher in the control group than in the DPP-4 inhibitor group (10.0% vs. 2.0%, *p* = 0.017). There was no significant difference in the prescribed medications related to the bone metabolism, such as calcium, vitamin D, and bisphosphonate, between the two groups.

### 3.2. Changes in BMD and TBS during Follow-Up

At baseline, there were no significant differences in the BMD and TBS between the two groups (Table 1). Table 2 and Figure 1 summarize the changes in the BMD and TBS values during the 1-year follow-up. The lumbar spine BMD increased significantly in both groups (from 0.939 ± 0.181 g/cm^2^ to 0.958 ± 0.172 g/cm^2^ in the DPP-4 inhibitor group, *p* < 0.001 and from 0.927 ± 0.179 g/cm^2^ to 0.941 ± 0.182 g/cm^2^ in the control group, *p* = 0.036), indicating no significant differences in the delta change between these two groups (*p* = 0.576; Figure 1A). There was an increased but statistically insignificant TBS value between the baseline and follow-up DXA examinations in the DPP-4 group (from 1.227 ± 0.119 to 1.241 ± 0.121, *p* = 0.149). In contrast, the TBS value in the control group slightly decreased between the two DXA examinations (from 1.227 ± 0.114 to 1.215 ± 0.118, *p* = 0.095), These results indicated that there was a significant difference in the delta change in the TBS between the two groups (*p* = 0.030; Figure 1B).

### 3.3. Changes in Anthropometric and Laboratory Parameters

Table 3 shows the changes in the different variables measured between patients treated with or without DPP-4 inhibitors. Compared with the baseline, the serum calcium levels increased in the DPP-4 inhibitor group (from 9.4 ± 0.6 mg/dL to 9.6 ± 0.6 mg/dL, *p* = 0.036) but did not change in the control group (from 9.4 ± 0.7 mg/dL to 9.3 ± 0.7 mg/dL, *p* = 0.323), which led to a significant difference between the two groups (*p* = 0.032). The 25-hydroxy vitamin D3 also increased from 24.8 ± 14.7 ng/mL to 31.0 ± 11.2 ng/mL in the DPP-4 inhibitor group (*p* = 0.004), while, in the control group, only a slight increase was observed (from 30.5 ± 8.9 ng/mL to 32.0 ± 9.7 ng/mL, *p* = 0.471). There was no statistical significance between the two groups (*p* = 0.137), suggesting that there were no meaningful changes in the other variables between both groups.

### 3.4. Changes in Medications during the Follow-Up Period

Medications that were newly prescribed during the follow-up period are shown in Table 4. A higher number of patients started bisphosphonate in the control group than in the DPP-4 inhibitors group (16.0% vs. 6.0%, *p* = 0.040). Other medications including calcium and vitamin D were not different between these two groups.

## 4. Discussion

In this retrospective study, we investigated the effect of DPP-4 inhibitors on bone health in patients with T2DM. The BMD showed a significant increase over a one-year period both in the DPP-4 inhibitors and control treatment groups. However, it is noteworthy that increased TBS were observed only in patients treated with DPP-4 inhibitors, suggesting that DPP-4 inhibitors might have a beneficial effect on bone health in patients with T2DM.

Several possible mechanisms related to DPP-4 inhibitors on bone metabolism have been suggested in previous studies [15]. Negative effects on bone health in patients with T2DM are attributed to reduced bone formation via hyperglycemia and insulin resistance [16] DPP-4 inhibitors may affect bone health by improving hyperglycemia as antihyperglycemic agents that extend the half-life of incretin hormones such as glucagon like peptide-1 (GLP-1) and gastric inhibitory polypeptide (GIP) [17]. In experimental studies, GLP-1 induced osteoblast proliferation and apoptosis inhibition by binding to GLP-1 receptors expressed on the osteocyte cell surface [18,19]. Moreover, previous animal studies demonstrated that GLP-1 have a beneficial effect on BMD, bone strength, and bone architecture [20,21]. Similarly, GIP also showed a positive effect on bone health by affecting the GIP receptor expressed on osteoblast sand osteoclasts [22,23].

Despite this mechanistically supportive evidence, it is inconsistent with observations made in previous clinical studies. A meta-analysis of 28 randomized clinical trials demonstrated that treatment with DPP-4 inhibitors was significantly associated with a reduced fracture risk compared with the placebo or other antihyperglycemic agents [24]. In contrast, the results from another meta-analysis that included 51 randomized clinical trials showed no significant association with the fracture risk in patients treated with DPP-4 inhibitors when compared with the placebo or other antihyperglycemic drugs [10]. In a previous study performed in postmenopausal women, a treatment with sitagliptin for 12 weeks did not change the BMD, while bone turnover markers showed significant changes over the course of the study [8]. However, another study conducted in drug-naive T2DM showed that treatment with vildagliptin for one year did not affect bone turnover markers [25]. In this context, our results may indicate that DPP-4 inhibitors have a favorable effect on bone health in patients with T2DM.

In our study, the TBS significantly increased in patients treated with DPP-4 inhibitors, while the BMD increased in both the treatment and control groups. Generally, the BMD measurements by dual energy bone densitometry is used for the assessment of osteoporosis and risk of fracture. It is well-known that patients with T2DM have a higher risk for a fracture, even in patients with normal or increased BMD [11]. Therefore, there is some limitation in using BMD to assess the bone health in diabetic patients, and other alternative methods are needed for an accurate disease assessment. Bone quality is as important a parameter for bone strength as the bone mass. The TBS was recently introduced as a tool to measure bone strength by evaluating bone microstructure [13]. In recent studies, the TBS was shown to be a useful assessment tool for a fracture risk in diabetic patients [14,26]. However, there is a lack of studies that evaluate the effect of antihyperglycemic agents on TBS. One randomized controlled study demonstrated that TBS did not change in patients treated with metformin compared to the placebo [27]. Our study highlights an important association between DPP-4 inhibitors and bone strength when assessed by TBS.

Increases in the serum calcium and 25-hydroxy vitamin D3 levels were observed in the DPP-4 inhibitors group despite no significant changes in medication use between the two groups. It is possible that these differences might be influenced by drug compliance, such as calcium and vitamin D, because almost all the patients were already taking these medications at the baseline. However, a previous study conducted in 295 patients with T2DM reported that their serum 25-hydroxy vitamin D3 levels were significantly higher in patients treated with DPP-4 inhibitors than those treated with other antihyperglycemic mediations [28]. This could be one possible explanation for the results reported in the present study.

To the best of our knowledge, this is the first study to evaluate the effect of DPP-4 inhibitors on bone health assessments using TBS in patients with T2DM. However, the results of our study also had limitations, including that the retrospective study design had a relatively small number of subjects. It was possible that there are some confounding factors that can affect the results. In addition, the duration of treatment with medications such as antihyperglycemic agents and those related to osteoporosis among subjects was not standardized; therefore, it is not certain whether the results of this study were only due to the effect of the drug or whether there were other factors that affected these outcomes. A prospective randomized controlled study to confirm the present results and basic research to characterize the underlining mechanism is still needed.

In conclusion, the present study showed an increment in TBS in patients with T2DM treated with DPP-4 inhibitors, while there was no change in the control group, showing that there was a significant difference between the two groups. Additionally, serum calcium and 25-hydroxy vitamin D3 can increase in DPP-4 inhibitor users, suggesting that DPP-4 inhibitors may have a protective effect on bone health in patients with T2DM.

## Figures and Tables

**Figure 1 jcm-10-04775-f001:**
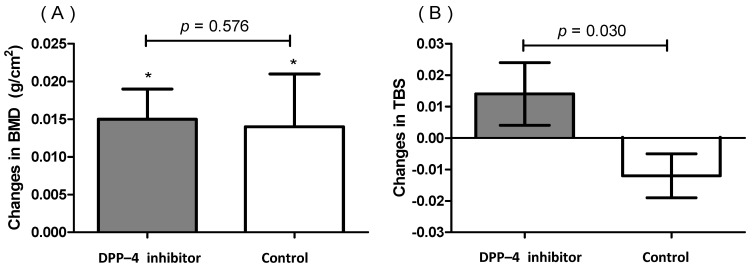
Changes in (**A**) the BMD and (**B**) TBS during the follow-up period after the DPP-4 inhibitor treatment. Data are expressed as the mean ± SE. * *p* < 0.05 by the paired *t-*test between the values recorded at the baseline and follow-up. *p*-values were calculated using the Student’s *t*-test for changes between the two groups.

**Table 1 jcm-10-04775-t001:** Baseline characteristics of the study population.

	DPP-4 Inhibitor (*n* = 100)	Control (*n* = 100)	*p*-Value
Age (years)	67.7 ± 10.5	68.1 ± 11.1	0.824
Sex, M/F (*n*)	10/90	12/88	0.822
Height (cm)	154.3 ± 7.6	154.8 ± 7.4	0.698
Weight (kg)	60.9 ± 11.7	58.5 ± 11.4	0.156
Body mass index (kg/m^2^)	25.5 ± 3.8	24.4 ± 4.4	0.087
Duration of DM (years)	9.4 ± 9.1	8.5 ± 9.1	0.551
Family history of DM, *n* (%)	35 (35.0)	17 (17.0)	0.006
Laboratory findings			
FPG (mg/dL)	139.9 ± 39.8	132.2 ± 36.8	0.454
HbA1c (%)	7.5 ± 1.4	6.7 ± 1.1	0.001
C-peptide (ng/mL)	2.1 ± 1.5	2.3 ± 1.4	0.616
Calcium (mg/dL)	9.4 ± 0.6	9.4 ± 0.7	0.373
Phosphorous (mg/dL)	3.7 ± 0.6	3.7 ± 0.7	0.614
BUN (mg/dL)	18.1 ± 17.5	19.2 ± 12.9	0.616
Creatinine (mg/dL)	0.80 ± 0.85	1.10 ± 1.24	0.055
Protein (g/dL)	6.9 ± 0.5	6.8 ± 0.6	0.127
Albumin (g/dL)	4.7 ± 3.9	4.6 ± 3.9	0.874
AST (IU/L)	25.0 ± 16.9	26.2 ± 14.5	0.581
ALT (IU/L)	24.0 ± 19.7	25.0 ± 22.0	0.757
ALP (IU/L)	72.6 ± 28.3	85.3 ± 56.5	0.049
Total cholesterol (mg/dL)	162.7 ± 43.8	176.6 ± 75.0	0.113
Triglycerides (mg/dL)	140.4 ± 57.4	147.0 ± 84.6	0.598
HDL-cholesterol (mg/dL)	50.3 ± 20.2	52.5 ± 16.5	0.528
LDL-cholesterol (mg/dL)	94.2 ± 38.6	105.6 ± 44.0	0.135
25-Hydroxyvitamin D (ng/mL)	24.0 ± 14.1	26.7 ± 10.9	0.399
Parathyroid hormone (pg/mL)	74.7 ± 88.4	63.3 ± 63.8	0.635
C-telopeptide (ng/mL)	0.356 ± 0.244	0.838 ± 1.393	0.176
Osteocalcin (ng/mL)	13.76 ± 6.06	40.36 ± 71.66	0.146
Lumbar spine BMD (g/cm^2^)	0.939 ± 0.181	0.927 ± 0.179	0.631
Lumbar spine TBS (unitless)	1.227 ± 0.119	1.227 ± 0.114	0.992
Medications			
Sulfonylurea	31 (31.0)	10 (10.0)	<0.001
Metformin	81 (81.0)	62 (62.0)	0.004
SGLT2 inhibitor	0 (0.0)	1 (1.0)	1.000
Insulin	2 (2.0)	10 (10.0)	0.017
Calcium	54 (54.0)	58 (58.0)	0.669
Vitamin D	70 (70.0)	79 (79.0)	0.194
Bisphosphonate	40 (40.0)	43 (43.0)	0.774
SERM	4 (4.0)	7 (7.0)	0.537
Parathyroid hormone	1 (1.0)	1 (1.0)	1.000
Denosumab	2 (2.0)	1 (1.0)	1.000

Data are expressed as the mean ± standard deviation (SD) or *n* (%). The *p*-values were calculated using the Student’s *t*-test for continuous data and chi-square test for categorical data. DM, diabetes mellitus; FPG, fasting plasma glucose; HbA1c, hemoglobin A1c; BUN, blood urea nitrogen; AST, aspartate aminotransferase; ALT, alanine aminotransferase; ALP, alkaline phosphatase HDL, high-density lipoprotein; LDL, low-density lipoprotein; BMD, bone mineral density; TBS, trabecular bone score; SGLT2, sodium glucose cotransporter 2; SERM, selective estrogen receptor modulator.

**Table 2 jcm-10-04775-t002:** Changes in the BMD and TBS during the follow-up period.

	DPP-4 Inhibitor			Control			^†^*p*-Value	
	Baseline	after 1 Year	*p* Value	Baseline	after 1 Year	*p*-Value		
BMD (g/cm^2^)	0.939 ± 0.181	0.958 ± 0.172	<0.001	0.927 ± 0.179	0.941 ± 0.182	0.036	0.576	
TBS	1.227 ± 0.119	1.241 ± 0.121	0.149	1.227 ± 0.114	1.215 ± 0.118	0.095	0.030	

Data are expressed as the mean ± SD. BMD, bone mineral density; TBS, trabecular bone score. *p*-values were calculated using a paired *t-*test between the values recorded at the baseline and after one-year follow-up. ^†^
*p*-values were calculated using the Student’s *t-*test for changes between the two groups.

**Table 3 jcm-10-04775-t003:** Changes in the anthropometric variables and biomarkers.

	DPP-4 Inhibitor			Control			
	Baseline	after 1 Year	*p*-Value	Baseline	after 1 Year	*p*-Value	^†^*p*-Value
Body mass index (kg/m^2^)	25.5 ± 3.8	25.2 ± 4.1	0.050	24.4 ± 4.4	24.5 ± 4.6	0.850	0.263
FPG (mg/dL)	137.7 ± 40.4	133.0 ± 44.5	0.364	136.2 ± 37.0	129.5 ± 41.3	0.323	0.844
HbA1c (%)	7.5 ± 1.4	7.2 ± 1.2	0.067	6.8 ± 1.2	6.7 ± 1.0	0.604	0.422
Calcium (mg/dL)	9.4 ± 0.6	9.6 ± 0.6	0.036	9.4 ± 0.7	9.3 ± 0.7	0.323	0.032
Phosphorous (mg/dL)	3.7 ± 0.6	3.6 ± 0.6	0.497	3.7 ± 0.7	3.7 ± 0.7	0.735	0.837
BUN (mg/dL)	18.1 ± 17.5	17.1 ± 7.1	0.583	19.4 ± 13.0	20.6 ± 14.2	0.448	0.370
Creatinine (mg/dL)	0.80 ± 0.85	0.82 ± 0.76	0.312	1.12 ± 1.27	1.24 ± 1.50	0.092	0.195
AST (IU/L)	25.0 ± 16.9	24.1 ± 15.5	0.591	26.5 ± 14.7	25.9 ± 23.6	0.812	0.897
ALT (IU/L)	24.0 ± 19.7	22.6 ± 18.3	0.426	25.4 ± 22.3	23.5 ± 16.4	0.405	0.846
ALP (IU/L)	72.5 ± 28.3	68.1 ± 23.1	0.050	83.3 ± 52.0	80.8 ± 51.1	0.552	0.676
Total cholesterol (mg/dL)	162.7 ± 43.8	160.6 ± 39.8	0.625	177.7 ± 76.2	166.3 ± 37.8	0.153	0.290
Triglycerides (mg/dL)	138.0 ± 54.6	139.4 ± 69.1	0.838	148.5 ± 85.1	156.6 ± 94.9	0.488	0.607
HDL-cholesterol (mg/dL)	50.7 ± 20.2	51.4 ± 14.7	0.700	53.5 ± 15.3	55.3 ± 15.5	0.340	0.720
LDL-cholesterol (mg/dL)	89.9 ± 34.0	87.8 ± 27.3	0.601	107.9 ± 42.0	102.3 ± 28.7	0.355	0.611
25-Hydroxyvitamin D (ng/mL)	24.8 ± 14.7	31.0 ± 11.2	0.004	30.5 ± 8.9	32.0 ± 9.7	0.471	0.137
Parathyroid hormone (pg/mL)	78.2 ± 94.2	51.1 ± 26.9	0.113	64.8 ± 80.1	68.2 ± 132.0	0.853	0.291
C-telopeptide (ng/mL)	0.356 ± 0.244	0.351 ± 0.246	0.933	0.476 ± 0.485	0.638 ± 1.005	0.415	0.292
Osteocalcin (ng/mL)	14.09 ± 6.40	14.60 ± 10.77	0.847	18.76 ± 14.61	15.45 ± 22.58	0.538	0.468

Data are expressed as the mean ± SD. FPG, fasting plasma glucose; HbA1c, hemoglobin A1c; BUN, blood urea nitrogen; AST, aspartate aminotransferase; ALT, alanine aminotransferase; ALP, alkaline phosphatase HDL, high-density lipoprotein; LDL, low-density lipoprotein. *p*-value was calculated using a paired *t-*test between the values recorded at the baseline and after 1-year follow-up. ^†^
*p*-values were calculated using the Student’s *t-*test for changes between the two groups.

**Table 4 jcm-10-04775-t004:** Changes in the medications (newly started).

	DPP-4 Inhibitor (*n* = 100)		Control (*n* = 100)		*p*-Value
Sulfonylurea	7	(7.0)	2	(2.0)	0.170
Metformin	9	(9.0)	3	(3.0)	0.134
Insulin	3	(3.0)	2	(2.0)	1.000
Calcium	13	(13.0)	11	(11.0)	0.828
Vitamin D	14	(14.0)	15	(15.0)	1.000
Bisphosphonate	6	(6.0)	16	(16.0)	0.040
SERM	2	(2.0)	5	(5.0)	0.445
Parathyroid hormone	0	(0.0)	1	(1.0)	1.000
Denosumab	1	(1.0)	1	(1.0)	1.000

Data are expressed as *n* (%). The *p*-values were calculated using the chi-square test. SERM, selective estrogen receptor modulator.

## Data Availability

The datasets used and analyzed in the present study could be made available from the corresponding author (endoann@daum.net) upon reasonable request.

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
