# Peer review of "Effect of Dipeptidyl Peptidase-4 Inhibitors on Bone Health in Patients with Type 2 Diabetes Mellitus"

_jcm, 2021, doi:10.3390/jcm10204775_

Round 1
Reviewer 1 Report
This manuscript reports on measures of bone health (most notably the trabecular bone score) in individuals with type 2 diabetes based on exposure to the anti-glycemic medication of DPP-4 inhibitor. The authors demonstrate that the DPP-4 group experienced increased trabecular bone score compared to controls who experienced a decreased trabecular bone score, which was indicative of a statistically significant difference in delta between the two groups. The authors also report an increase in serum calcium and 25-OH vitamin D in the DPP-4 group. Studies of this nature are important to better delineate novel manners to assess fracture risk and bone health in individuals with T2D and which treatment modalities and factors may be risk factors or protective. Strengths of this study include: important and relevant topic, novel use of trabecular bone score in this population, excellent discussion of potential mechanisms relating DPP-4 inhibitors and bone metabolism in the discussion section, and a well written and concise manuscript. However, a few items should be clarified to improve the manuscript.
1) The biggest item apparent after reading the manuscript is the potential for confounders other then DPP-4 inhibitors that are contributing or causing the difference in trabecular bone score noted between the two groups, especially in the setting that HgbA1c and several diabetes related medications (insulin, metformin, etc.) were significantly different between the two groups. If it is not possible to control for at least a few of the mentioned potential confounders in any of the analysis, some of the language should be softened that implies more than a possible association. For example, in the discussion, page 9 lines 228-229, “Our study highlights an important effect of DPP-4 inhibitors on bone strength”. This wording implies causation by using the word effect. It should be changed to something along the lines of, “…highlights an important association between DPP-4 inhibitors and bone strength”. This type of language should be used consistently throughout the discussion when describing the results and conclusions that can be made. This potential weakness of confounders is mentioned in the discussion section, which is important.
2) In the introduction, it is importantly mentioned that BMD in individuals with T2D is higher than those without diabetes. As the authors transition to talking about utilizing trabecular bone score, is there is any literature to cite that reports on differences in TBS between those with diabetes and those without? If such a reference exists, it would be helpful to include it with a statement in the introduction. If this is unknown, a quick mention in the discussion that the differences in TBS between those with and without diabetes is unknown would be helpful for the reader.
3) At the end of the introduction, page 2, line 65: “aim of this study was to evaluate the beneficial effect of DPP-4 inhibitors on bone health…” Given this is likely the hypothesis of the study, it may be helpful to state “potential benefit” or similar wording instead.
4) Small grammatical aspects to consider:
- Page 1, line 36: “common complications are related to include vascular conditions such as…” reads awkwardly. Consider, “common complications include vascular conditions such as…”
- Page 1, line 40: “…osteoporotic fractures are significantly associated with morbidity and mortality”, the word “significantly” in a manuscript of this nature lends the reader to think of “statistically significant” instead of the other (presumed intended) definition. Perhaps use a word like “strongly associated” or “importantly associated”.
- In section 2.3 of the materials and methods, repeat a similar line twice: “The same DXA instrument was used for both measurements” and “the same DXA instrument was used to take two measurements”; can either delete one or provide additional clarification if they are referring to two different measurements.
Author Response
[Response to Reviewer #1]
Comments and Suggestions for Authors
This manuscript reports on measures of bone health (most notably the trabecular bone score) in individuals with type 2 diabetes based on exposure to the anti-glycemic medication of DPP-4 inhibitor. The authors demonstrate that the DPP-4 group experienced increased trabecular bone score compared to controls who experienced a decreased trabecular bone score, which was indicative of a statistically significant difference in delta between the two groups. The authors also report an increase in serum calcium and 25-OH vitamin D in the DPP-4 group. Studies of this nature are important to better delineate novel manners to assess fracture risk and bone health in individuals with T2D and which treatment modalities and factors may be risk factors or protective. Strengths of this study include: important and relevant topic, novel use of trabecular bone score in this population, excellent discussion of potential mechanisms relating DPP-4 inhibitors and bone metabolism in the discussion section, and a well written and concise manuscript. However, a few items should be clarified to improve the manuscript.
1) The biggest item apparent after reading the manuscript is the potential for confounders other then DPP-4 inhibitors that are contributing or causing the difference in trabecular bone score noted between the two groups, especially in the setting that HgbA1c and several diabetes related medications (insulin, metformin, etc.) were significantly different between the two groups. If it is not possible to control for at least a few of the mentioned potential confounders in any of the analysis, some of the language should be softened that implies more than a possible association. For example, in the discussion, page 9 lines 228-229, “Our study highlights an important effect of DPP-4 inhibitors on bone strength”. This wording implies causation by using the word effect. It should be changed to something along the lines of, “…highlights an important association between DPP-4 inhibitors and bone strength”. This type of language should be used consistently throughout the discussion when describing the results and conclusions that can be made. This potential weakness of confounders is mentioned in the discussion section, which is important.
à We appreciate the reviewer’s careful assessment of our manuscript. We totally agree with reviewer’s opinion. According the reviewer’s suggestion, we have modified in the revised manuscript.
[Revised] p. 9, lines 229–231 and 244–245 in the revised manuscript
Our study highlights an important association between DPP-4 inhibitors and bone strength when assessed by TBS.
It was possible that there are some confounding factors that can affect the results.
2) In the introduction, it is importantly mentioned that BMD in individuals with T2D is higher than those without diabetes. As the authors transition to talking about utilizing trabecular bone score, is there is any literature to cite that reports on differences in TBS between those with diabetes and those without? If such a reference exists, it would be helpful to include it with a statement in the introduction. If this is unknown, a quick mention in the discussion that the differences in TBS between those with and without diabetes is unknown would be helpful for the reader.
à Thank you for your precise and insightful comment. There is a study shown that TBS was lower in patients with T2DM than non-diabetic individuals. We have added this contents in the revised manuscript.
[Revised] p. 2, lines 63–65 in the revised manuscript
In another study, TBS was lower in patients with T2DM than normal controls, indicating that TBS can be a useful predictor for bone health in patients with T2DM.
[References for revision]
Ho-Pham, L.T.; Nguyen T.V. Association between Trabecular Bone Score and Type 2 Diabetes: A Quantitative Update of Evidence. Osteoporos Int 2019, 30, 2079-85.
3) At the end of the introduction, page 2, line 65: “aim of this study was to evaluate the beneficial effect of DPP-4 inhibitors on bone health…” Given this is likely the hypothesis of the study, it may be helpful to state “potential benefit” or similar wording instead.
à We thank the reviewer for this valuable comment. We have modified the sentence in the revised manuscript according to the reviewer’s opinion.
[Revised] p. 2, lines 67–68 in the revised manuscript
The aim of this study was to evaluate the potential benefits of DPP-4 inhibitors on bone health using TBS in patients with T2DM.
4) Small grammatical aspects to consider:
Page 1, line 36: “common complications are related to include vascular conditions such as…” reads awkwardly. Consider, “common complications include vascular conditions such as…”
Page 1, line 40: “…osteoporotic fractures are significantly associated with morbidity and mortality”, the word “significantly” in a manuscript of this nature lends the reader to think of “statistically significant” instead of the other (presumed intended) definition. Perhaps use a word like “strongly associated” or “importantly associated”.
In section 2.3 of the materials and methods, repeat a similar line twice: “The same DXA instrument was used for both measurements” and “the same DXA instrument was used to take two measurements”; can either delete one or provide additional clarification if they are referring to two different measurements.
à We appreciate the reviewer’s careful review. We have modified this contents in the revised manuscript.

Reviewer 2 Report
The authors investigated the association between bone health and DPP4 inhibitor treatment in type 2 diabetics.
Asking this question is interesting, timely, important.
The design of the study is appropriate to answer this question, however, as the study was retrospective in nature,
which brings with it the disadvantages of this type of study:
it was not possible to form homogeneous patient groups, which is certainly a limiting factor.
Although the results are modest in absolute terms, the direction of change is favorable in the DPP4-inhibitory branch and unfavorable in the control branch,
and is therefore remarkable.
The authors modestly note that a randomized controlled trial would be required to answer the question correctly.
For the sake of correctness, I propose to expand the paragraph on the limiting factors of the study.
I would like to know what they think about the difference in ALP, creatinine and HbA1c levels.
In my opinion, in the conclusion chapter, a few sentences should be written about any significant and nearly significant differences between the two groups.
Overall, the writing is eye-catching, I recommend it publishing after minor changes.
Author Response
[Response to Reviewer #2]
Comments and Suggestions for Authors
The authors investigated the association between bone health and DPP4 inhibitor treatment in type 2 diabetics. Asking this question is interesting, timely, important. The design of the study is appropriate to answer this question, however, as the study was retrospective in nature, which brings with it the disadvantages of this type of study: it was not possible to form homogeneous patient groups, which is certainly a limiting factor. Although the results are modest in absolute terms, the direction of change is favorable in the DPP4-inhibitory branch and unfavorable in the control branch, and is therefore remarkable. The authors modestly note that a randomized controlled trial would be required to answer the question correctly. For the sake of correctness, I propose to expand the paragraph on the limiting factors of the study.
I would like to know what they think about the difference in ALP, creatinine and HbA1c levels.
In my opinion, in the conclusion chapter, a few sentences should be written about any significant and nearly significant differences between the two groups.
Overall, the writing is eye-catching, I recommend it publishing after minor changes.
à We appreciate the reviewer’s careful assessment of our manuscript. As pointed out by reviewer, there are some limitations due to retrospective design of our study. There might be some confounders to affect our results. We have added this content in the limitation section of the revised manuscript. With same context, differences in some laboratory findings at baseline did not control in this study. However, as shown in Table 3, there were no differences in those parameters. In addition, according to the reviewer’s suggestion, we have added sentences of our results in the conclusion section of the revised manuscript.
[Revised] p. 9, lines 244–245 and 251–253 in the revised manuscript
It was possible that there are some confounding factors that can affect the results.
In conclusion, the present study showed an increment in TBS in patients with T2DM treated with DPP-4 inhibitors, while there was no change in the control group, showing that there was a significant difference between the two groups.
